# Meta-Analysis: Urinary Calprotectin for Discrimination of Intrinsic and Prerenal Acute Kidney Injury

**DOI:** 10.3390/jcm8010074

**Published:** 2019-01-10

**Authors:** Jia-Jin Chen, Pei-Chun Fan, George Kou, Su-Wei Chang, Yi-Ting Chen, Cheng-Chia Lee, Chih-Hsiang Chang

**Affiliations:** 1Department of Nephrology, Kidney Research Center, Chang Gung Memorial Hospital, Taoyuan 333, Taiwan; raymond110234@hotmail.com (J.-J.C.); franwis1023@gmail.com (P.-C.F.); b92401107@gmail.com (G.K.); ytchen@mail.cgu.edu.tw (Y.-T.C.); chia7181@gmail.com (C.-C.L.); 2Graduate Institute of Clinical Medical Science, College of Medicine, Chang Gung University, Taoyuan 333, Taiwan; 3Clinical Informatics and Medical Statistics Research Center, College of Medicine, Chang Gung University, Taoyuan 333, Taiwan; shwchang@mail.cgu.edu.tw; 4Division of Allergy, Asthma, and Rheumatology, Department of Pediatrics, Chang Gung Memorial Hospital, Taoyuan 333, Taiwan; 5Department of Biomedical Sciences, College of Medicine, Chang Gung University, Taoyuan 333, Taiwan

**Keywords:** urine calprotectin, acute kidney injury, intrinsic renal injury

## Abstract

**Background:** Urinary calprotectin is a novel biomarker that distinguishes between intrinsic or prerenal acute kidney injury (AKI) in different studies. However, these studies were based on different populations and different AKI criteria. We evaluated the diagnostic accuracy of urinary calprotectin and compared its diagnostic performance in different AKI criteria and study populations. **Method:** In accordance with Preferred Reporting Items for Systematic Reviews and Meta-Analyses (PRISMA) guidelines, we searched PubMed, Embase, and the Cochrane database up to September 2018. The diagnostic performance of urinary calprotectin (sensitivity, specificity, predictive ratio, and cutoff point) was extracted and evaluated. **Result:** This study included six studies with a total of 502 patients. The pooled sensitivity and specificity were 0.90 and 0.93, respectively. The pooled positive likelihood ratio (LR) was 15.15, and the negative LR was 0.11. The symmetric summary receiver operating characteristic (symmetric SROC) with pooled diagnostic accuracy was 0.9667. The relative diagnostic odds ratio (RDOC) of the adult to pediatric population and RDOCs of different acute kidney injury criteria showed no significant difference in their diagnostic accuracy. **Conclusion:** Urinary calprotectin is a good diagnostic tool for the discrimination of intrinsic and prerenal AKI under careful inspection after exclusion of urinary tract infection and urogenital malignancies. Its performance is not affected by different AKI criteria and adult or pediatric populations.

## 1. Introduction

Acute kidney injury (AKI) is a common and widespread problem with high mortality and morbidity. Despite understanding the pathogenesis of different etiologies, traditional diagnosis markers (including serum creatinine and urine output) are not a real-time, not a sensitive and specific renal marker for early diagnosis and interventions, not based on acute kidney etiology, and the differentiation of prerenal injury and intrinsic kidney injury is difficult. There are numerous causes of AKI, which are most commonly classified as prerenal, intrinsic (intrarenal), or postrenal kidney injury. To date, many studies have revealed that neutrophil gelatinase-associated lipocalin (NGAL) has shown promising results in the early diagnosis of AKI [1,2,3,4], distinguishing between prerenal and intrinsic kidney injury ([5,6,7], and predicting the need for renal replacement therapy and prognosis. [8,9]. Urinary calprotectin is a heterodimer protein involved in the immune system [10] and plays a role in the AKI pathophysiology. Early studies have shown that the release of urinary calprotectin from neutrophil and renal tubular epithelial cells also produces calprotectin in response to injury [10,11,12]. Calprotectin has been demonstrated to be similar to NGAL as a diagnostic marker for early diagnosis and to make a different diagnosis of AKI etiology [5,13,14,15,16,17]. This biomarker, which can early detect acute kidney injury and distinguish between prerenal and intrinsic AKI, can facilitate intervention, reduce the time to initiate therapy, and reduce the number of unnecessary renal biopsies. Nevertheless, these studies used different AKI criteria and were based on different populations. Therefore, we conducted a systemic review and meta-analysis for evaluating the differential diagnostic accuracy of urinary calprotectin between prerenal and intrinsic kidney injury.

## 2. Methods

### 2.1. Literature Search

Our two investigators (J.-J.C., C.-H.C.) systematically and independently conducted a review of the published data in accordance with Preferred Reporting Items for the Systematic Reviews and Meta-Analyses (PRISMA) guideline. A computerized search of the electronic databases of Pubmed, Embase, and the Cochrane database was performed to identify all relevant English-language studies up to September 2018 using the keywords and medical subject heading (MeSH) term: AKI, calprotectin, S100A8/A9 complex, and myeloid-related protein complex.

### 2.2. Study Selection

Two investigators independently determined the study eligibility based on an evaluation of the titles, abstracts, and subsequently, the full texts. Any difference in opinion regarding eligibility was resolved by consensus through discussion. Any article that was deemed potentially relevant was retrieved online for the full-text. Studies were included if they met the following criteria: full-length English original articles published and available, human studies, urinary calprotectin for distinguishing between intrinsic and pre-renal AKI, clear definition of AKI: (the Risk, Injury, Failure, Loss, End-Stage Kidney Disease (RIFLE), AKI Network (AKIN), Kidney Disease Improving Global Outcomes (KDIGO), or pediatric RIFLE criteria (pRIFLE)) and reported the definition/clinical criteria of intrinsic or prerenal AKI. Studies were excluded according to the following criteria: (1) focusing on chronic kidney disease, (2) duplicate cohort, (3) non-original studies (such as reviews, commentaries, letters), (4) studies with insufficient information, (5) studies that were not based on urinary calprotectin level, (6) studies with no reported intrinsic or prerenal AKI. Review articles or meta-analysis were not included in the analysis; however, their citations and references were searched for additional relevant studies. Full search strategies are available in Appendix A.

### 2.3. Data Extraction

Two investigators (J.-J.C., C.-H.C.) independently extracted the relevant information from each study. Data elements related to the study level characteristics included first author, year of publication, study location, study design, definition of AKI, sample processing, method of storage, calprotectin measurement method, and test kit, see Table 1. As for patient characteristics, data included gender, age, diabetes, hypertension, urinary tract infection (UTI), creatinine on admission, creatinine prior to admission, C-reactive protein, urinary creatinine, urinary calprotectin, and urinary calprotectin to creatinine ratio and are summarized in Table 2. Items related to the diagnostic test performance were also extracted, including cutoff points based on the Youden index, sensitivity, specificity, and the number of intrinsic and prerenal kidney injuries, see Table 3. 

### 2.4. Outcome Measures

The diagnostic criteria of AKI were different in the six enrolled studies. Four of which (Heller, 2011; Seibert, 2013; Seibert, 2016; Basiratnia, 2017) used AKIN criteria [18]. One (Chang, 2015) used KDIGO AKI criteria [19]. One (Westhoff, 2016) used the pRIFLE criteria modified by Ackan-Arikan et al. [20]. Two of which (Westhoff, 2016; Basiratnia, 2017) were pediatric population studies.

The reference test for differentiating intrinsic or prerenal acute kidney diagnosis was based on clinical criteria as mentioned below (most studies used predefined criteria). The histologic diagnosis of hepato-renal syndrome or cardio-renal syndrome was considered the golden standard. The response to volume repletion (return of creatinine to baseline within 48 to 72 h) was considered an obligatory diagnostic criterion for prerenal kidney injury. Other findings for the diagnosis of prerenal kidney injury included compatible history (dehydration, fluid loss, heart failure, liver cirrhosis), compatible physical examination (low blood pressure, low jugular pulse, tachycardia, orthostatic blood pressure changes, poor skin turgor), and compatible urine analysis (no proteinuria, no hematuria). UTI was classified as an intrinsic kidney injury in three enrolled studies (Heller, 2011; Seibert, 2013; Seibert, 2016).

### 2.5. Risk of Bias

We used the Quality Assessment of Diagnostic Accuracy Studies 2 (QUADAS-2) tool and Review Manager version 5.3 to assess the quality of the included studies [21]. The QUADAS-2 score is based on four domains (patient selection, index test, reference standard, and flow and timing) to judge the risk of bias. Each study was reviewed independently by J.-J.C., C.-H.C., and rated as high, low, or of unclear risk for all four domains. The judgment principle of “applicability” was the same as the bias section, but there were no signaling questions. Disagreements between the two reviewers were solved by consensus through discussion. If the answer to all signaling questions in each domain is “yes”, the domain is considered as low risk. If any signaling question is answered “no”, the domain is considered as having a high risk of bias.

### 2.6. Statistical Analysis

True positive (TP), true negative (TN), false positive (FP), and false negative (FN) rates for each study were calculated according to the reported sensitivity, specificity, and patient number of prerenal and intrinsic AKI. Based on these data, the positive likelihood ratio (+LR), negative likelihood ratio (−LR), and diagnostic odds ratio (DOR) could be obtained for each study. The summary measures were calculated using a random effects model (DerSimonian and Laird method). To assess the diagnostic performance of urinary calprotectin in predicting intrinsic AKI in AKI patients, a symmetric summary receiver operating characteristic (symmetric SROC) curve was constructed based on TP and FP rates. The threshold effect was detected using the Spearman correlation coefficient between the logit of sensitivity and logit of ‘1−specificity’, where a non-significant threshold effect was warranted before performing further subgroup analysis or meta-regression [22]. The degree of heterogeneity among studies was evaluated using the *I*^2^ index, with <25%, 25%–50%, and >50% indicating mild, moderate, and high heterogeneity, respectively. Likelihood ratios indicate that the accuracy of a particular test would be more accurate for patients with a disease than for subjects without disease. Two variables (adult vs. pediatric; AKI criteria) were performed as moderators in the meta-regression analyses to explore possible sources of heterogeneity. A sensitivity analysis was done to exclude patients with a UTI. All analyses were conducted by Meta-DiSc (version 1.4) software [23]. A two-sided *p* value of <0.05 was considered statistically significant.

## 3. Results

### 3.1. Literature Search

The initial search retrieved 83 records. After excluding duplicated articles and removing irrelevant articles, the remaining 30 articles were screened based on the title and abstract. Ten potentially relevant articles were identified and full-text articles were downloaded and accessed for eligibility. Of these 10 articles, one of which [16] was suspected of using a duplicate cohort to another study [15], two of which reported no data of intrinsic and prerenal AKI, and one of which had no data on urinary calprotectin. Finally, six studies were included in this meta-analytic study, see Figure 1.

### 3.2. Risk of Bias

With the QUADAS-2 tool, some study characteristics that might increase the risk of bias were identified. Domain 1 of QUADAS-2 focused on patient selection. Four of the included studies were based on an adult population and two on a pediatric population. One of the studies (Seibert, 2016) selected a population that was not a consecutive or random sample of patients but rather focused on post-kidney-transplant adults. Another study (Chang, 2015) selected a narrow spectrum population in the coronary care unit (CCU). Domain three addresses aspects of the reference standard. Inconsistent standard criteria of AKI (KDIGO, AKIN, or pRIFLE) and a lack of pathological evidence of intrinsic kidney injury were found in all studies. In addition, most studies used the clinical observation of a rapid decrease in serum creatinine with convergence to the baseline within 72 h after fluid repletion to diagnose prerenal AKI, except for one study (Basiratnia, 2017) that used 48 h as a time interval. Because there was one study (Seibert, 2016) with an adult kidney transplant population and two others (Basiratnia, 2017, Westhoff 2016) with pediatric populations, the answer regarding the applicability of the patient selection of these three studies was considered to be unclear. We summarized the risk of bias data for all the included studies in Figure 2.

### 3.3. Study Characteristics 

The characteristics of the six included studies are summarized in Table 1. Four of the studies were performed in Germany, one in Taiwan, and one in Iran. Sample sizes ranged from 53 to 152 patients. Two studies were conducted on pediatric populations, one on an adult kidney transplant population, and three on adult populations. Two studies excluded patients with UTI, two provided data for the entire cohort and data of excluded UTI patients, and three provided data of urinary calprotectin and normalization by urine creatinine. The optimal cutoffs were determined by the Youden index in three studies. These six studies adopted clinical diagnostic criteria for prerenal or intrinsic kidney injury, including rapid decreasing serum creatinine (Cr) (<72 h) after fluid repletion as prerenal AKI, physical examination finding, and urine examination. The detailed information of the reference test is described earlier in this article. 

### 3.4. Patient’s Characteristics 

A total of 502 patients were included in these six studies. All studies were single-center trials. Study populations included adult and children populations, kidney transplant populations, and CCU patients. The mean age of the four adult AKI studies was 68 years, and there were more males with prerenal acute kidney than males with an intrinsic kidney injury (*p* < 0.001). The prevalence of hypertension, diabetes mellitus, and UTI was higher in the intrinsic kidney group. The level of serum creatinine on admission or AKI diagnosis was not significantly different (*p* = 0.054). The C-reactive protein level was also not significantly different (*p* = 0.412). Not surprisingly, the urinary calprotectin and urinary calprotectin to creatinine ration were higher in patients with an intrinsic kidney injury. Two pediatric AKI studies had a mean age of 7.5 and 6.0 years in the prerenal and intrinsic kidney injury groups, respectively. The level of serum creatinine on admission or AKI diagnosis was significantly higher in the pediatric intrinsic kidney injury group (*p* < 0.001). Detailed information is summarized in Table 2. 

### 3.5. Urinary Calprotectin for Discriminating Prerenal and Intrinsic Acute Kidney Injuries

The diagnostic values, cutoffs, and key results are summarized in Table 3. The pooled sensitivity and specificity were 0.90 (95% CI: 0.87–0.93) and 0.93 (95% CI: 0.88–0.96), respectively. The pooled positive LR was 15.15 (95% CI: 4.45–51.55), and the negative LR was 0.11 (95% CI: 0.06–0.20), as shown in Figure 3. The symmetric SROC with pooled diagnostic accuracy was 0.9667, see Figure 4. The heterogeneity of the aforementioned four pooled indices was moderate to high (*I*^2^ ranged from 55% to 68.9%).

By using normalization according to urine creatinine, the data of three studies were pooled. The pooled sensitivity and specificity were 0.93 (95% CI: 0.87–0.97) and 0.95 (95% CI: 0.88–0.98), respectively. The pooled positive LR was 14.75 (95% CI: 5.54–39.3), and the negative LR was 0.08 (95% CI: 0.04–0.18), see Appendix A. The SROC with pooled diagnostic accuracy was 0.9840, see the Appendix A.

### 3.6. Subgroup Analysis

Due to the moderate to high heterogeneity, several study characteristics (population age and criteria of AKI) were used to explore the sources of heterogeneity. The analysis of the diagnosis threshold was performed with Spearman rank correlation (ρ = −0.429; *p* = 0.397), indicating no threshold effect and allowing for further subgroup analysis. The relative diagnostic odds ratio (RDOC) of the adult population relative to the pediatric population was 2.48 (95% CI: 0.01–737.91), indicating no significant difference in the diagnostic accuracy between adult and pediatric cohorts. The RDOCs of AKIN and KDIGO (both relative to RIFLE) were 25.13 (95% CI: 0.04–15927.64) and 5.38 (95% CI: 0.01–4757.39), respectively, indicating no significant difference in the diagnostic accuracy under different criteria of AKI (data not shown). 

### 3.7. Sensitivity Analyses 

There were three studies that provided data after excluding patients with a UTI. The pooled sensitivity and specificity of these two studies were 0.92 (95% CI: 0.85–0.96) and 0.98 (95% CI: 0.92–1.00), respectively. The pooled positive LR was 31.95 (95% CI: 9.40–108.54), and the negative LR was 0.10 (95% CI: 0.05–0.17), see the Appendix A. The symmetric SROC with pooled diagnostic accuracy was 0.9995, see the Appendix A.

## 4. Discussion

Calprotectin is a heterodimer protein (S100A8/S100A9) that plays a role in the innate immune system, acute kidney pathophysiology, and kidney repair processes as described below. Our findings can be summarized in the following points: (1) Urinary calprotectin is a good marker for differentiation of intrinsic and prerenal AKI; (2) the diagnostic performance of urinary calprotectin is not significantly different in different acute kidney diagnostic criteria and in adult or pediatric populations. 

The urinary calprotectin is higher in intrinsic kidney injury than prerenal kidney injury. It may be reasonable to conclude that urinary calprotectin is a good diagnostic test in the discrimination of an intrinsic kidney injury with a pooled diagnostic accuracy of symmetric SROC of 0.9667. It has been noted in earlier studies that calprotectin is released from the immune system cells (neutrophils and to lesser degree monocytes) and renal collecting duct epithelial cells [10,11,25,26]. It has also been demonstrated that renal tubular epithelial cells produce calprotectin in response to unilateral ureteral obstruction [11]. Calprotectin also increases expression after ischemia-reperfusion injury and plays a role in M2 macrophage-mediated renal repair [12]. It acts as a danger-associated molecular pattern protein that activates toll-like receptor 4 (TLR4). The available immunostainings of the clinical studies suggest that inflammatory infiltration rather than the tubular epithelial cells is the major source of urinary calprotectin in AKI [13,27]. Therefore, different etiologies of an intrinsic kidney injury which involved calprotectin, neutrophils infiltration, and TLR4 are expected to have higher urinary calprotectin. For example, in the leading causes of intrinsic kidney injury, renal epithelial tubular damage and inflammatory renal disease (including glomerulonephritis, tubular-interstitial nephritis and vasculitis, pyelonephritis) can lead to higher levels of urinary calprotectin. In contrast, in prerenal AKI, there is a functional deficit leading to low levels of urinary calprotectin. Elevated urinary calprotectin has been described in different diseases such as urinary bladder malignancies [28]. Gastroenterologists also used fecal calprotectin to distinguish between function disorder (irritable bowel syndrome) and inflammatory bowel diseases [29,30].

Heller (2011) has indicated that a UTI has a higher urinary calprotectin level than other intrinsic kidney injury causes. Pyuria is a potential confounder because it increases the calprotectin level in the urine, independent of renal function. Three above-mentioned studies (Heller, 2011; Seibert, 2013; Seibert, 2016) enrolled a UTI population as having intrinsic kidney injury. Three of the six enrolled studies (Heller, 2011; Chang 2015; Basiratnia, 2017) reported population or subgroup data showing an accuracy after exclusion of UTI and the symmetric SROC of pooled diagnostic accuracy was 0.9995. This might suggest that the diagnostic value of calprotectin is better if UTI can be excluded before examination. 

Our research also supports the notion that the diagnostic accuracy of urinary calprotectin does not differ from different AKI criteria. The current AKI criteria are based on serum creatinine and urine output. It is widely noted that serum creatinine is not only a delayed but also a functional marker, rather than a damage marker to kidney injury. The novel biomarker was elevated earlier than serum Cr, and in a previous human renal ischemia-reperfusion study [26], calprotectin even increased earlier than NGAL (2 h and 8 h after injury, respectively). This may be an explanation for why we found that the accuracy of urinary calprotectin is not interfered by different AKI criteria. 

Calprotectin has several characteristics that make it a promising novel marker and even a troponin for nephrologists [31]. First, as mentioned above, it rises earlier than NGAL. Second, according to Azimi [31], calprotectin combined with serum endocan may further differentiate pure tubular injury from glomerular-tubular injury. In addition, calprotectin has been reported to be associated with mortality and can predict the progression of kidney disease. In an AKI pediatric population, Westhoff et al. concluded that urinary calprotectin can predict the 30-day mortality and the need for renal replacement therapy [16]. Another kidney transplantation adult population study conducted by Tepel et al. revealed that urinary calprotectin levels on day 1 after operation predicted allograft injury and renal function decline after 1 month, 6 months, and 12 months after surgery [32]. 

The first limitation concerns the moderate to high heterogeneity of enrolled studies due to different study populations, even in adult patients (cardiac care unit and kidney transplant populations). As with other similar AKI biomarker systemic studies [33], different acute kidney definitions are also sources of heterogeneity. The second limitation is that our enrolled studies are all published online, the data may represent an optimistic estimate. In addition, few studies have addressed the role of calprotectin so far and only six articles were enrolled in our studies. Furthermore, to date, there is no clinical golden standard for the diagnosis of intrinsic AKI, and current studies are all based on history, clinical, and physical examination criteria. This may result in the misclassification of kidney injury etiology. Urogenital malignancies and UTI may increase urinary calprotectin concentrations independent of acute kidney injury. The careful inspection for urogenital malignancies and UTI is warranted before clinical application.

## 5. Conclusion

In conclusion, early diagnosis of acute kidney injury is of great significance to clinical practice and guides further therapy. Our study demonstrated that urinary calprotectin is a good diagnostic marker for discriminating intrinsic and prerenal AKI in adult or pediatric populations, and its performance was not interfered by different AKI criteria. Further large, multicenter trials may be needed to clarify and identify the possible role of urinary calprotectin in different populations. More efforts on developing biomarkers to guide therapy or treatment protocol and more rapid and accurate etiology diagnosis for AKI are still needed before before the troponin of nephrologist coming true.

## Figures and Tables

**Figure 1 jcm-08-00074-f001:**
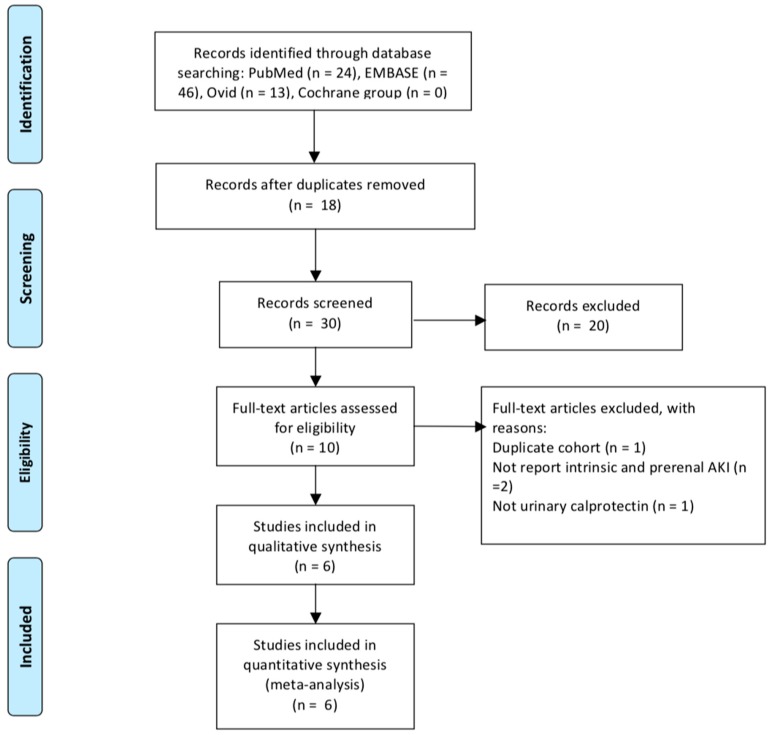
Preferred Reporting Items for Systematic Reviews and Meta-Analyses (PRISMA) flowchart.

**Figure 2 jcm-08-00074-f002:**
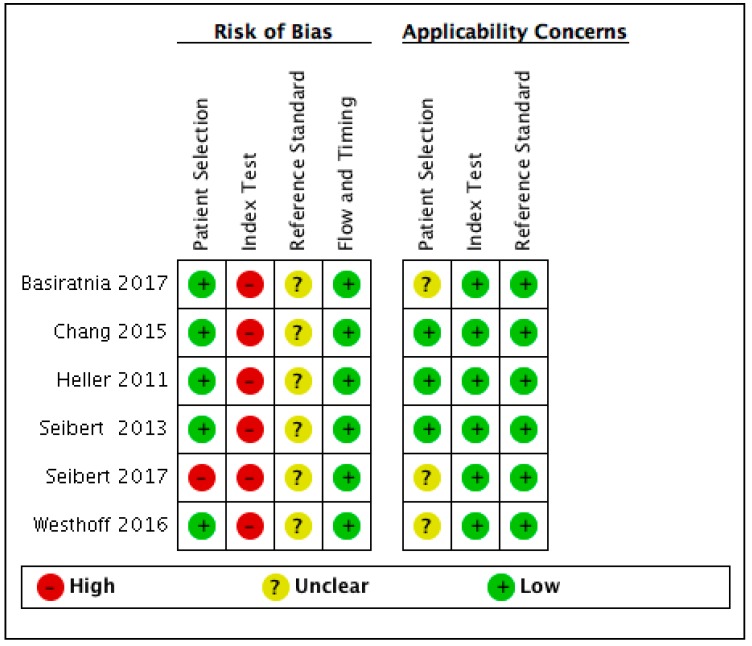
Summary of risk of bias and applicability concerns.

**Figure 3 jcm-08-00074-f003:**
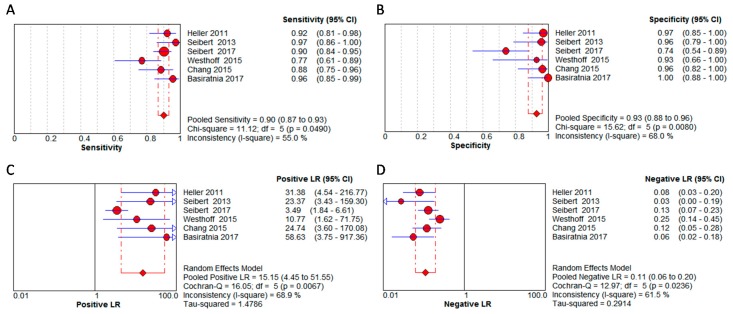
Diagnostic performance of urinary calprotectin on discriminating between intrinsic acute kidney injuries and prerenal acute kidney injury.

**Figure 4 jcm-08-00074-f004:**
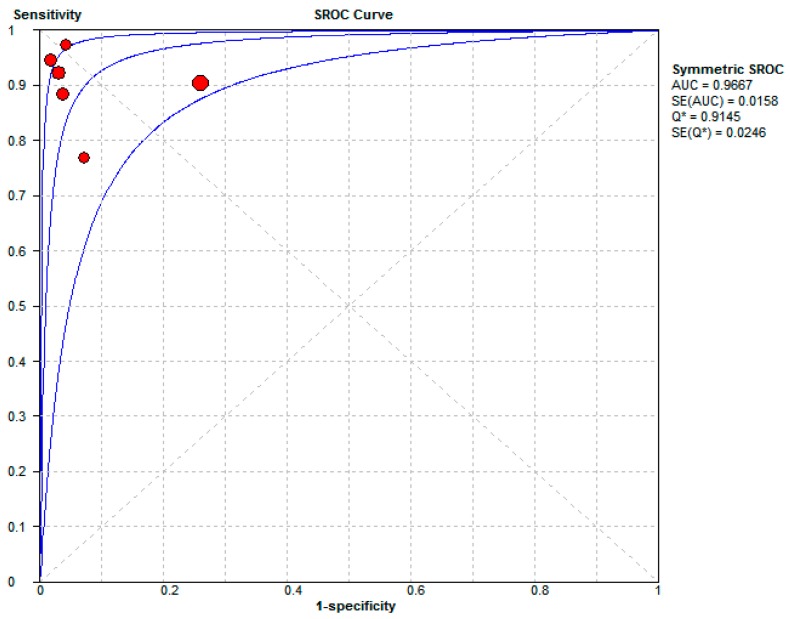
Symmetric summary receiver operating characteristic (symmetric SROC) according to the cutoffs of the six studies. Abbreviation: SROC, summary receiving operating characteristics; AUC, area under the curve; SE, standard error.

**Table 1 jcm-08-00074-t001:** The characteristics of the six included studies.

Study/year	Location	Design	AKI Criteria	Population	Sample Time	Storage	Assay	Test Kit
Basiratnia/2017 [24]	Iran	PC	AKIN	Pediatric	Immediately at diagnosis of AKI	−20 °C, no centrifugation	ELISA	PhiCal^®^ Calprotectin, catalogue number K 6928; Immundiagnostik AG, Bensheim, Germany
Chang/2015 [17]	Taiwan	PC	KDIGO	Adult CCU	Immediately at admission	−80 °C, centrifugation	ELISA	R&D Systems, DLCN20, McKinley Place NE Minneapolis; MPLS, USA and Phi Cal^®^ Calprotectin, K 6935; and Immundiagnostik AG, Bensheim, Germany
Heller/2011 [13]	Germany	PC	AKIN	Adult	Within 3 days	−20 °C, no centrifugation	ELISA	PhiCal^®^ Calprotectin, catalog number K 6935; Immundiagnostik AG, Bensheim, Germany
Seibert/2013 [5]	Germany	PC	AKIN	Adult	NR	−20 °C, no centrifugation	ELISA	PhiCal^®^ Calprotectin, catalog number K 6935; Immundiagnostik AG, Bensheim, Germany
Seibert/2017 [14]	Germany	PC	AKIN	Adult transplant	At admission or on clinics	−20 °C, no centrifugation	ELISA	PhiCal^®^ Calprotectin, catalogue number K 6928; Immundiagnostik AG, Bensheim, Germany
Westhoff/2016 [15]	Germany	PC	pRIFLE	Pediatric	Immediately at diagnosis or after admission with AKI.	−80 °C, centrifugation	ELISA	PhiCal^®^ Calprotectin; Immundiagnostik AG, Bensheim, Germany

Abbreviation: AKI (acute kidney injury), AKIN (Acute Kidney Injury Network), CCU (coronary care unit), ELISA (enzyme linked immunosorbent assay), KDIGO (Kidney Disease Global Outcomes), NR (not reported), pRIFLE (pediatric Risk, Injury, Failure, Loss of kidney function and End stage kidney disease), PC (prospective cohort).

**Table 2 jcm-08-00074-t002:** Patient characteristics based on available data.

Variable	Adult			Pediatric		
	Prerenal (*n* = 116)	Intrinsic (*n* = 258)	*p*	Prerenal (*n* = 44)	Intrinsic (*n* = 84)	*p*
Male (%)	73.3	49.2	<0.001	54.6	40.5	<0.001
Age (years)	68 (66, 68)	68 (58, 71)	0.007	7.5 (2.6, 7.5)	6.0 (0.6, 6.0)	<0.001
Hypertension (%)	80.2	82.6	<0.001	NA	NA	NA
Diabetes (%)	28.5	30.6	<0.001	NA	NA	NA
Urinary tract infection (%)	0	41.1	<0.001	NA	NA	NA
Creatinine on admission or diagnosis (mg/dL)	3.1 (2.6, 4.4)	3.4 (3.1, 4.1)	0.054	1.1 (0.9, 1.1)	1.8 (1.8, 1.9)	<0.001
Creatinine at baseline (mg/dL)	1.4 (1.4, 1.7)	1.4 (1.4, 1.9)	0.023	NA	NA	NA
CRP (mg/dL)	5.2 (3.5, 5.3)	5.1 (0.7, 6.7)	0.412	NA	NA	NA
Urine creatinine (g/L)	0.7 (0.7, 0.8)	0.6 (0.5, 0.6)	<0.001	NA	NA	NA
Urinary calprotectin (ng/mL)	54 (28, 385)	1955 (1955, 2405)	<0.001	29 (19, 29)	1240 (427, 1240)	<0.001
Urinary calprotectin (ng/mL)/Cr (g/L) ratio	57 (52, 310)	2775 (2775, 3698)	<0.001	NA	NA	NA

CRP, C-reactive protein; NA, not applicable; Cr, creatinine; continuous variable was presented as median and interquartile range.

**Table 3 jcm-08-00074-t003:** Summary of diagnostic performance of the six included studies.

Study/Year	Sample Size	Event (Prerenal/Intrinsic)	Cufoff (ng/mL)	Sensitivity	Specificity	PPV	NPV
Basiratnia 2017	75	30/45	230	96.7	96.7	97.7	96.7
Chang 2015	74	31/43	314.6	88.4	96	NR	NR
Heller 2011	86	34/52	300	92.3	97.1	98	89.2
Seibert 2013	62	24/38	600	97.4	95.8	97.4	95.8
Seibert 2017	152	27/125	134.5	90.4	74.1	NR	NR
Westhoff 2016	53	14/39	76	77	93	97	60

PPV, positive predicted value; NPV, negative predicted value; NR, not reported.

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
