# Peer review of "Meta-Analysis: Urinary Calprotectin for Discrimination of Intrinsic and Prerenal Acute Kidney Injury"

_jcm, 2019, doi:10.3390/jcm8010074_

Round 1
Reviewer 1 Report
The diagnostic use of urinary calprotectin for the discrimination of prerenal and intrinsic acute kidney injury was first reported in 2011. Since that time there have been several studies on this approach in different patient populations and by different authors. Chen et al. present a first meta-analysis on this topic summarizing the current data.
AKI is a widespread entity with a high impact on morbidity and mortality. Its incidence has increased substantially in the past decades. Thus, the topic of the manuscript is of high clinical relevance. The approach to differentiate two major forms of AKI by calprotectin is novel and there has been no metaanalysis on the available clinical studies so far.
The authors correctly follow the PRISMA guideline. Data extraction, choice of outcome measures and statistical analysis are correct. The manuscript is well written. Tables and figures adequately illustrate the study populations and pooled results.
Abstract: In the conclusion, “identification” has to be exchanged by “discrimination” of intrinsic and prerenal AKI, since the low calprotectin concentation in prerenal AKI is similar to healthy controls.
In the Discussion Chen et al. present additional data on calprotectin’s origin, pathophysiology, dynamics, and future perspective. The clinical limitations of this new biomarker, however, should be presented more precisely in a separate paragraph. The two major limitations are urogenital malignancies and urinary tract infections, which increase urinary calprotectin concentrations independent of acute kidney injury. The correct interpretation of measurement findings thereby necessitates an exclusion of UTI. These two limitations should be included in the abstract as well.
Page 13, ll. 251-254: The authors should add the fact that the available immunostainings of the clinical studies suggest that rather the inflammatory infiltration than the tubular epithelial cells is the major source of urinary calprotectin in AKI.
Reference 16 is cited incorrectly. Please change citation style and correct name of first author Basiratnia.
Author Response
Dear section editor, assigned editor, and reviewers,
We are very pleased that you offered us an opportunity to revise our work entitled “Meta-analysis: urinary calprotectin for discrimination of intrinsic kidney injury and prerenal acute kidney injury” for Journal of Clinical Medicine. The comments and insights were a tremendous help to us. In the revised manuscript, in accordance with the valuable suggestions of the reviewers, we have made some modifications. The major concerns of the reviewers have been fully addressed, and the entire manuscript has been carefully revised. All the revised words and sentences were highlighted by Track Changes. We hope that the revised manuscript will fulfill the requirements of reviewers and you will judge the revised manuscript to be suitable for publication in Journal of Clinical Medicine.
Yours sincerely,
First author: Jia-Jin Chen
Corresponding author: Chih-Hsiang Chang
Point 1: Abstract: In the conclusion, “identification” has to be exchanged by “discrimination” of intrinsic and prerenal AKI, since the low calprotectin concentration in prerenal AKI is similar to healthy controls.
Response 1: We thank you for your kind words and your precious time to review our article. According to your suggestion, we have modified our abstract (line 30).
Point 2: The clinical limitations of this new biomarker, however, should be presented more precisely in a separate paragraph. The two major limitations are urogenital malignancies and urinary tract infections, which increase urinary calprotectin concentrations independent of acute kidney injury. The correct interpretation of measurement findings thereby necessitates an exclusion of UTI. These two limitations should be included in the abstract as well.
Response 2: Thanks for your kindly reminder. We have revised the abstract (line 31) and limitation (line 299).
Point 3: The authors should add the fact that the available immunostainings of the clinical studies suggest that rather the inflammatory infiltration than the tubular epithelial cells is the major source of urinary calprotectin in AKI.
Response 3: Thanks for your valuable suggestions. We modified the section of discussion (line 255).
Point 4: Reference 16 is cited incorrectly. Please change citation style and correct name of first author Basiratnia.
Response 4: We thank you for your inspection. We have changed our citation (line 425).

Reviewer 2 Report
In the present study, Jia-Jin Chen et al. carried out a systematic review and meta-analysis of urinary calprotectin for discrimination of intrinsic and prerenal AKI. They selected 6 studies and concluded that urinary calprotectin has a good specificity and sensitivity for their discrimination. Technically, this is a straight-forward work.
[Major Point]
I agree that diagnostic performance of urinary calprotectin to distinguish prerenal and intrinsic AKI is quite good in patients without urinary tract infection (UTI)/ pyuria/leukocyturia. However, patients with potential prerenal AKI having UTI have been carefully removed from most studies included in this review. Therefore, a limitation should be clearly described in the abstract that usefulness of urinary calprotection in patients with UTI has not been established.
[Minor point]
Page 5. Sample sizes among 6 selected studies were not 53-280.
Author Response
Dear section editor, assigned editor, and reviewers,
We are very pleased that you offered us an opportunity to revise our work entitled “Meta-analysis: urinary calprotectin for discrimination of intrinsic kidney injury and prerenal acute kidney injury” for Journal of Clinical Medicine. The comments and insights were a tremendous help to us. In the revised manuscript, in accordance with the valuable suggestions of the reviewers, we have made some modifications. The major concerns of the reviewers have been fully addressed, and the entire manuscript has been carefully revised. All the revised words and sentences were highlighted by Track Changes. We hope that the revised manuscript will fulfill the requirements of reviewers and you will judge the revised manuscript to be suitable for publication in Journal of Clinical Medicine.
Yours sincerely,
First author: Jia-Jin Chen
Corresponding author: Chih-Hsiang Chang
Review 2
Point 1: I agree that diagnostic performance of urinary calprotectin to distinguish prerenal and intrinsic AKI is quite good in patients without urinary tract infection (UTI)/ pyuria/leukocyturia. However, patients with potential prerenal AKI having UTI have been carefully removed from most studies included in this review. Therefore, a limitation should be clearly described in the abstract that usefulness of urinary calprotection in patients with UTI has not been established.
Response 1: Thank you for your valuable review and comments. We have added this clinical limitation to our abstract (line 31)
Point 2: Page 5. Sample sizes among 6 selected studies were not 53-280.
Response 2: We thank the reviewer for reminding us of this mistake. We have changed it (53 to 152) (line 162)
